# Bioinformatic Analysis for Mucoepidermoid and Adenoid Cystic Carcinoma of Therapeutic Targets

**DOI:** 10.3390/vaccines10091557

**Published:** 2022-09-19

**Authors:** Carla Monserrat Ramírez-Martínez, Luis Fernando Jacinto-Alemán, Luis Pablo Cruz-Hervert, Javier Portilla-Robertson, Elba Rosa Leyva-Huerta

**Affiliations:** 1Oral Medicine and Pathology Department, Postgraduate and Research Division, Dentistry School, National Autonomous University of Mexico, Mexico City 04510, Mexico; 2Orthodontic Department, Postgraduate and Research Division, Dentistry School, National Autonomous University of Mexico, Mexico City 04510, Mexico

**Keywords:** adenoid cystic carcinoma, mucoepidermoid carcinoma, salivary gland, gene expression and bioinformatics

## Abstract

Salivary gland neoplasms are a heterogeneous neoplasm group, including mucoepidermoid carcinoma (MECa), adenoid cystic carcinoma (AdCC), and many others. Objective: We aimed to identify new critical genes of MECa and AdCC using bioinformatics analysis. Methods: Gene expression profile of GSE153283 was analyzed by the GEO2R online tool to use the DAVID software for their subsequent enrichment. Protein–protein interactions (PPI) were visualized using String. Cytoscape with MCODE plugin followed by Kaplan–Meier online for overall survival analysis were performed. Results: 97 upregulated genes were identified for MECa and 86 for AdCC. PPI analysis revealed 22 genes for MECa and 63 for AdCC that were validated by Kaplan–Meier that showed FN1 and SPP1 for MECa, and EGF and ERBB2 for AdCC as more significant candidate genes for each neoplasm. Conclusion: With bioinformatics methods, we identify upregulated genes in MECa and AdCC. The resulting candidate genes as possible therapeutic targets were FN1, SPP1, EGF, and ERBB2, and all those genes had been tested as a target in other neoplasm kinds but not salivary gland neoplasm. The bioinformatic evidence is a solid strategy to select them for more extensive research with clinical impact.

## 1. Introduction

Salivary gland neoplasms can originate from structures that comprise the parenchymal or glandular stromal, constituting a group of morphologically diverse tumors [1]. Its incidence is 2.5 cases per 100,000 persons per year [2]. The biomolecular mechanisms involved in their etiology are unknown. Recent studies focused on investigating mutations in genes that may be associated with the origin of these tumors. According to the 2017 WHO classification, these neoplasms are categorized according to their biological behavior in 11 benign and 20 malignant entities, of which mucoepidermoid carcinoma (MECa) and adenoid cystic carcinoma (AdCC) are the most frequent malignant neoplasms, representing approximately 35% and 21.9% of cases, respectively [3,4].

MECa appears in wide age distribution, with a peak incidence in the second decade of life. It affects both major and minor salivary glands, with a predilection for the female sex [3]. Clinically and histopathologically, it shows various degrees of malignancy, ranging from non-aggressive or low-grade neoplasms and intermediate-grade neoplasms to aggressive or high-grade neoplasms. Cellular diversity and histological patterns of salivary gland neoplasms are frequently misdiagnosed as lymphomatous papillary cystadenoma, adenoid cystic carcinoma, and squamous cell carcinoma [5,6].

The etiopathogenesis of MECa remains uncertain, and recent research aimed to elucidate their development. Researchers have suggested changes in the DNA sequence, such as alterations in TP53, mutations in the transcription factor POU6F2 involved in cell differentiation, and the translocation t (11; 19) (q21; p13), which gives rise to the MECT1-MAML2 fusion oncoprotein [7,8,9,10].

Concerning AdCC, the mean age of patients diagnosed was 57 years old, without gender predilection. It could affect both the major and minor salivary glands [11]. Clinically and histopathologically, it shows three degrees of malignancy described in 1984 by Szanto et al. [12]. Low-grade neoplasms are those with a tubular pattern, intermediate-grade neoplasms are those with a cribriform pattern, and high-grade neoplasms are those with a solid component. Less than 30% of AdCC have a great affinity for perineural invasion and presenting distant metastases, which is why they are considered aggressive neoplasms. Their five-year survival represents 55% to 89%, which decreases at 15 years from 20 to 40% [11]. Their etiopathogenesis has not been fully elucidated. However, in more than 60% of cases, the AdCC has been associated with the chromosomal translocation t (6; 9) (q22–23; p23–24), which gives rise to the MYB-NFIB fusion oncoprotein [13].

Although these alterations continue to be studied, they cannot explain the development of MECa and AdCC. This study aimed to identify new target genes using bioinformatics analysis for the prognosis of both malignant salivary gland neoplasms.

## 2. Materials and Methods

### 2.1. Microarray Data Information

NCBI-GEO is regarded as a free public microarray/gene profile database, and we obtained the gene expression profile of GSE153283 in primary salivary gland carcinomas and normal salivary gland tissues. Microarray data were all on account of the Nano Stringn Counter human PanCancer Pathways Panel, which included 14 MEC and 11 AdCC, each with their respective normal tissue.

### 2.2. Data Processing of Differential Expressed Genes (DEGs)

Candidate differentially expressed genes (DEGs) for MECa and AdCC with their normal specimens were identified via the GEO2R online tool with [logFC] > 1, and adjusted *p*-value < 0.05. DEGs with log FC > 1 were considered as an upregulated gene.

### 2.3. Functional Enrichment Analyses

The >1 FC gene list was submitted to DAVID 6.8 (available online: https://david.ncifcrf.gov, accessed on 31 May 2022) [14] to analyze the functions of the DEGs by Gene Ontology (GO) with the biological process, molecular function and cellular component; and Kyoto Encyclopedia of Genes and Genomes (KEGG) pathway enrichment. Statistical significance was set at *p* < 0.05. A Venn diagram was made for GO and KEGG of both neoplasms and determined which functions were similar for both.

### 2.4. Protein–Protein Interaction (PPI)

The Search Tool for the Retrieval of Interacting Genes (STRING; version 11.0, available online: https://string-db.org/, accessed on 31 May 2022) [15] database was employed to predict the protein–protein interaction (PPI) networks of the DEGs obtained. Next, the Cytoscape software was used to analyze the interaction with a combined score of >0.4 (Cytoscape; version 3.8.2, available online: http://cytoscape.org, accessed on 31 May 2022) [16]. The plugin molecular complex detection (MCODE) was used to detect the most significant module in the PPI networks with the MCODE score of degree cutoff = 2, node score cutoff = 0.2, k-core = 2, and max depth = 100.

### 2.5. Selection and Analyses of Hub Genes

For the selection of the hub genes, those clustered with MCODE score ≥ 6 were selected. Then, the effect of the hub genes on overall survival and disease-free survival was analyzed using the Kaplan–Meier plotter (KM plotter, available online: http://kmplot.com/analysis, accessed on 31 May 2022) by adjusting the follow-up threshold to 60 months [17]. To mimic the behavior of both neoplasm to the maximum extent concerning the candidate genes, both analyses were adjusted for head-neck squamous cell carcinoma (HNSCC).

## 3. Results

We identified 97 and 86 upregulated genes for MECa for AdCC, respectively, from an analysis of 32,000 gene sets. Both gene lists were analyzed by DAVID software, and the results of GO analysis for MECs indicated that the biological process was particularly enriched in extracellular matrix organization, collagen fibril organization, and collagen catabolic process; the molecular function was enriched in extracellular matrix structural constituent, identical protein binding, and integrin binding; and the cellular component was significantly enriched in extracellular region, space, and exosome. The AdCC showed that biological process was enriched in the positive regulation of the transcription from RNA polymerase II promoter, signal transduction, and positive regulation of transcription of the DNA template; molecular function was enriched in growth factor activity, protein binding, and Ras guanyl-nucleotide exchange factor activity; and the cellular component was significantly enriched in the plasma membrane, extracellular region, and membrane raft (Table 1). Five terms were matched when the common functions were analyzed—three cellular components (extracellular region, extracellular space, and extracellular exosome) and two molecular functions (Wnt-protein binding and growth factor activity).

KEGG analysis data (Table 2) showed that the upregulated genes were particularly enriched in ECM-receptor interaction, PI3K-Akt signaling pathway, and Focal adhesion for mucoepidermoid carcinoma; for adenoid cystic carcinoma, we observed enriched in pathways in cancer, the Ras signaling pathway, and transcriptional misregulation in cancer. The matched pathways observed for both neoplasms were the PI3K-Akt signaling pathway, pathways in cancer, and the Wnt signaling pathway; however, the genes displayed differed for each neoplasm.

### 3.1. PPI Network and Modular Analysis

A total of 22 DEGs, including 21 nodes and 70 edges for MECa, and 63 DEGs, with 25 nodes and 23 edges for AdCC, were imported into the PPI network complex. We then applied Cytoscape MCODE for further analysis. The results revealed 11 nodes and 45 edges for mucoepidermoid carcinoma and 8 nodes and 13 edges for adenoid cystic carcinoma, showing 10 clustered genes for mucoepidermoid and 6 clustered genes for adenoid cystic carcinoma (Figure 1 and Table 3).

### 3.2. Analysis of Core Genes Using a Kaplan–Meier Plotter

A Kaplan–Meier plotter was used to identify the survival data for the clustered genes. Only FN1 and SPP1 for MECa and EGF and ERBB2 for AdCC were significantly associated with poor survival (Figure 2).

## 4. Discussion

Cancer is a multistep process involving alterations or changes in the transcriptional activity of genes associated with many cellular processes for tumor development, including proliferation, senescence, and metastasis. Salivary gland neoplasms represent a significant challenge because their biological diversity leads to unpredictable treatment responses. Nowadays, it is not possible to improve worldwide the five-year survival rates of these lesions, which are generally reported between 60–80%, but can decrease to 37% in high-grade tumors [18,19,20].

MECa is the most common malignant salivary gland tumor, followed by AdCC. Tumor stage and grade have been used as survival predictors. It has been considered that low-grade MECa arises more often in minor salivary glands, and high-grade MEC arises more often in major salivary glands (frequently in the parotid gland) [21].

The chromosomal aberration due to CRTC1/3-MAML2 fusion is the most reported genetic alteration, whose fusion protein activates the transcription of cAMP target genes; however, this translocation act as a potential primary driver mutation. Germinal and somatic mutation, copy number variations, and gain or loss of different genes have been associated with CRTC1/3-MAML2 [22]. The AdCC is a slow-growing but mortal salivary gland neoplasm. Similar to MECa, the main genomic alteration is gene fusion (MYB-NFIB fusion is present in 29% to 86% of neoplasms). Applying bioinformatics methods on microarray profile datasets represents an important strategy to identify more useful therapeutic or prognostic biomarkers of salivary glands carcinomas. We obtained 97 and 86 upregulated genes for MECa and AdCC, respectively. The GO function and KEGG pathway analysis showed a heterogeneous expression pattern, with similarities in gene functions and pathways; however, the genes displayed in them were different. For this reason, the PPI network was constructed individually for each neoplasm. The genes that resulted from KM plotting validation were FN1 and SPP1 for MECa and EGF and ERBB2 for AdCC.

In recent years, high-throughput molecular biology techniques, such as high-throughput sequencing, rna-seq, gene microarray, or proteome analysis, have been used to search new markers to determine prognosis, histological lineage, or therapeutic targets. The information obtained from thousands of candidate genes or biomarkers has required the use of different bioinformatic tools, which can handle this large amount of information to be able to classify it, understand it, and give it clinical utility. Bioinformatic studies and their verification through molecular assays, such as PCR, Western blot, immunohistochemistry, or other bioinformatic databases, are a prerequisite to estimating the robustness and give us evidence or possible applications of differentially expressed candidate genes [23,24,25,26]. In this study, we use our analysis algorithm based on the genome-proteome-clinical utility premise; that is, we first explore results through GO and KEGG, and then look for their validated protein interaction, selecting only the genes with the highest correlation (clustered). Finally, we estimate its clinical usefulness based on one of the most important parameters, the five-year survival.

The FN1 is a glycoprotein that usually exists as a dimer. It has plasma FN, synthesized by liver hepatocytes, or cellular FN1, produced by fibroblasts, chondrocytes, myocytes, and synovial cells, that could assemble into insoluble fibrils. The FN interacts with other extracellular matrix (ECM) proteins, growth factors, glycosaminoglycans, cell surface receptors, and other molecules. These interactions are fundamental to inducing specific cell functions, such as differentiation and epithelial-mesenchymal transition. Fibril assembly is often upregulated during cancer [27]. In cancer, FN1 is expressed by cancer-associated fibroblasts and by the cancer cells. Leivo I et al. showed that salivary glands, including MECa, present overexpression of several genes, including FN1. Their analysis suggests that FN1 overexpression could be related to cell adhesion or shape. However, it is essential to consider that histopathological diversity and severity degree could be related to clinical behavior differences [28]. Many studies have postulated that ECM targeting is a possible alternative to cancer treatment. Inhibition of ECM components, remodeling enzymes, blockers for cell surface receptors as integrins that bind FN1, or targeting cancer-associated fibroblasts are other alternatives for cancer treatment [29]. These targets could be explored for MECa treatment.

Osteopontin (OPN) is a phosphoglycoprotein with many reported functions expressed by osteoclasts, osteoblasts, neurons, epithelial cells, T, B, NK, NK T, myeloid, and innate lymphoid cells. Similar to FN1, in cancer, OPN is overexpressed by tumoral parenchymal and stromal cells, and it has been implicated in invasion, metastasis, and treatment resistance. Its expression is associated with poor prognosis in glioma, melanoma, hepatocellular, prostate, lung, breast, colorectal, ovarian, and head and neck cancer [30]. It has been shown that OPN acts as a negative regulator of T cell activation and promotes the recruitment of macrophages. It has been proposed that these macrophages promote tumor growth by COX-2 through the promotion of angiogenesis and migration of cancer cells. Recently, immunotherapy to OPN in cancer has been proposed to be an alternative by neutralizing mAbs [31]. It has been reported that in salivary gland tumors, particularly in MECa, OPN preserves their overexpressed pattern compared with normal salivary gland tissue [32,33]. However, Fok et al. reported that OPN expression correlated with histological grade, which is an essential feature, since it reinforces that OPN can be used as a therapeutic target.

Epidermal growth factor (EGF) is secreted by ectodermic cells, monocytes, kidneys, and duodenal glands. It stimulates epithelial cell growth by binding their transmembrane receptor kinases to promote cell proliferation, survival, adhesion, migration, and differentiation. The EGF receptor (EGFR) family consists of four transmembrane receptors, including EGFR (HER1/erbB-1), HER2 (erbB-2/neu), HER3 (erbB-3), and HER4 (erbB-4). HER1 and HER2 are overexpressed in breast, non–small-cell lung, head and neck, and colon cancers, which is related to poor prognosis. The activation of ligand-receptor EGF-EGFR complex has been related to EMT and MMP9 in head and neck carcinoma and salivary adenoid cystic carcinoma [34,35,36]. Considering that this complex is a more accessible target, employing monoclonal antibodies is a recurrent strategy. Huang et al. reported that nimotuzumab, a humanized neutralizing G1 monoclonal antibody, can cause cell cycle arrest by suppressing in vitro proliferation of AdCC cells, which represents an important feature because the inhibition of these neoplasm cells could be translated to reduce or eliminate some clinical features of AdCC as metastasizes to lungs, bone, liver or brain, or recurrence. Trastuzumab and Pertuzumab are monoclonal antibodies against HER2, another member of the EGFR receptors family. HER2 (erbB-2/neu) is a tyrosine kinase that activates MAPK and PI3K pathways to cell growth and differentiation. In tumors, it is possible to observe gene amplification or overexpression that could be related to more aggressive and metastatic behavior, decreasing the survival expectation of a patient with many carcinomas, including AdCC [37].

An important variable to consider is activating one erbb member pathway could generate resistance in another [38]. Therefore, inhibiting the receptors may not be sufficient, and inhibiting the ligand is required. Considering that EGF is one of the essential ligands for EGFR, their concentration in human serum is variable. A high EGF amount plus EGFR overexpression could create conditions for tumor growth, even without specific EGFR mutations. Crombet et al. have proposed that targeting EGF is necessary for a complete treatment [39]. It has been reported that the most common genetic alteration in MECa and ACC is the presence of gene fusions [23,24] and that these fusions can even be used to subclassify the neoplasm, as well as from a molecular point of view be responsible for common gene alterations in carcinogenesis of salivary gland neoplasms. Although the bioinformatic approach can give greater order and understanding to a complex and uncertain panorama such as MECa and ACC, it is necessary to try to validate these results to achieve better results or applications. Chen Z et al. [24], through their bioinformatic analysis, have reported that their candidate genes may be the target of combined therapy against EGFR through drugs already approved by the FDA. These data, which coincide with our findings on EGF and erbb2, fulfill this therapeutic possibility, which provides a significant advance on the possible path that can be taken in the treatment of MECa and ACC.

## 5. Conclusions

Our bioinformatic analysis identified two hub genes (FN1 and SPP1) for MECa and two (EGF and ERBB2) for AdCC. The results supported in bioinformatic platforms showed that these genes play critical roles in the pathogenesis, progression, and prognosis of both salivary gland neoplasms. It is necessary to consider them as therapeutic targets to identify how we can affect MECa and AdCC in a specific way as tumor growth or metastasis; however, these results provide new and useful information for testing the new biomarkers.

## Figures and Tables

**Figure 1 vaccines-10-01557-f001:**
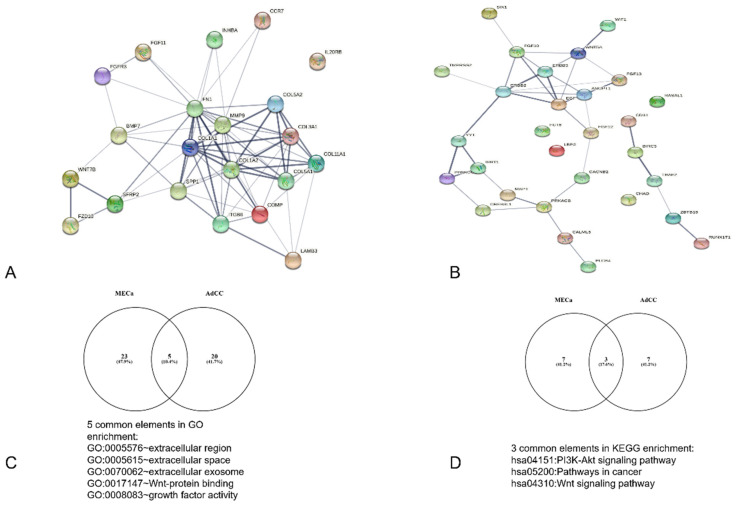
Analysis of the PPI interaction and sharedGO and KEGG interactions between MECa and AdCC. (**A**) PPI of MECa with central interactions with FN1, (**B**) PPI of ACC with principal interaction for EGF, (**C**,**D**) analysis of common shared elements of bioinformatic analysis.

**Figure 2 vaccines-10-01557-f002:**
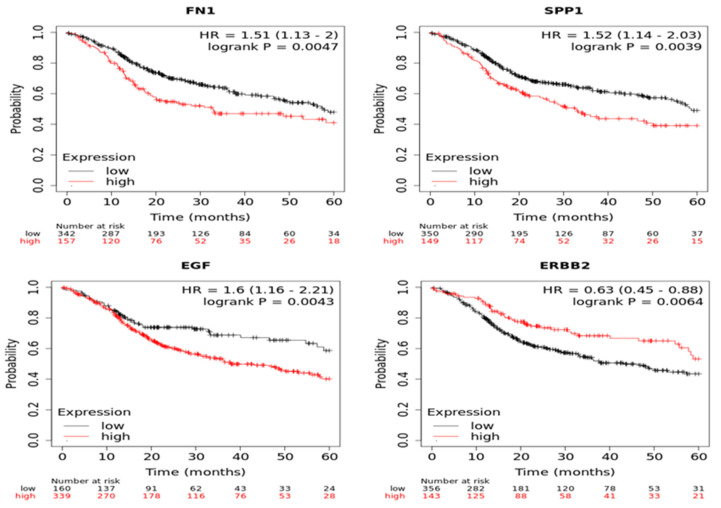
Analysis of core genes using Kaplan-Meier plotter FN1 and SPP1 for MECa and EGF and ERBB2 for AdCC.

**Table 1 vaccines-10-01557-t001:** GO analysis of differentially expressed genes associated with Mucoepidermoid and Adenoid cystic carcinoma.

Salivary Gland Carcinoma	Ontology	Term	Count	*p*-Value	FDR
Mucoepidermoid					
	BP	GO:0030198~extracellular matrix organization	11	6.34 × 10^−14^	2.71 × 10^−11^
	BP	GO:0030199~collagen fibril organization	7	1.37 × 10^−11^	2.92 × 10^−9^
	BP	GO:0030574~collagen catabolic process	7	3.08 × 10^−10^	4.38 × 10^−8^
	BP	GO:0007155~cell adhesion	7	3.57 × 10^−5^	0.00213038
	BP	GO:0071230~cellular response to amino acid stimulus	4	3.99 × 10^−5^	0.00213038
	BP	GO:0022617~extracellular matrix disassembly	4	1.68 × 10^−4^	0.00798976
	BP	GO:0050900~leukocyte migration	4	6.77 × 10^−4^	0.02892512
	BP	GO:0007267~cell-cell signaling	4	0.00547321	0.12983679
	BP	GO:0007275~multicellular organism development	4	0.03705893	0.43750278
	BP	GO:0045669~positive regulation of osteoblast differentiation	3	0.00329429	0.1004757
	CC	GO:0005576~extracellular region	19	7.88 × 10^−15^	4.10 × 10^−13^
	CC	GO:0005615~extracellular space	12	3.51 × 10^−7^	3.04 × 10^−6^
	CC	GO:0070062~extracellular exosome	10	0.00704296	0.04069265
	CC	GO:0031012~extracellular matrix	9	2.59 × 10^−9^	6.73 × 10^−8^
	CC	GO:0005578~proteinaceous extracellular matrix	8	3.84 × 10^−8^	6.66 × 10^−7^
	CC	GO:0005788~endoplasmic reticulum lumen	7	1.45 × 10^−7^	1.51 × 10^−6^
	CC	GO:0005581~collagen trimer	6	1.16 × 10^−7^	1.50 × 10^−6^
	CC	GO:0009986~cell surface	4	0.03330161	0.15742581
	MF	GO:0005201~extracellular matrix structural constituent	7	3.96 × 10^−10^	2.37 × 10^−8^
	MF	GO:0042802~identical protein binding	6	0.00356876	0.02379175
	MF	GO:0005178~integrin binding	5	1.36 × 10^−5^	2.73 × 10^−4^
	MF	GO:0048407~platelet-derived growth factor binding	4	4.14 × 10^−7^	1.24 × 10^−5^
	MF	GO:0008201~heparin binding	4	0.00146113	0.0144694
	MF	GO:0042813~Wnt-activated receptor activity	3	4.40 × 10^−4^	0.00659668
	MF	GO:0017147~Wnt-protein binding	3	8.78 × 10^−4^	0.01054057
	MF	GO:0046332~SMAD binding	3	0.0016881	0.0144694
	MF	GO:0005518~collagen binding	3	0.00326051	0.02379175
	MF	GO:0008083~growth factor activity	3	0.02199846	0.13199073
Adenoid cystic					
	BP	GO:0045944~positive regulation of transcription from RNA polymerase II promoter	7	0.00328478	0.65695556
	BP	GO:0007165~signal transduction	7	0.00748663	0.85056216
	BP	GO:0045893~positive regulation of transcription, DNA-templated	6	0.00102624	0.30787087
	BP	GO:0000165~MAPK cascade	5	6.61 × 10^−4^	0.30787087
	BP	GO:0000187~activation of MAPK activity	3	0.01183287	0.85056216
	BP	GO:0071222~cellular response to lipopolysaccharide	3	0.01312904	0.85056216
	BP	GO:0043524~negative regulation of neuron apoptotic process	3	0.01761926	0.85056216
	BP	GO:0006898~receptor-mediated endocytosis	3	0.03332279	1
	BP	GO:0001525~angiogenesis	3	0.04630619	1
	BP	GO:0014842~regulation of skeletal muscle satellite cell proliferation	2	0.00925562	0.85056216
	CC	GO:0005886~plasma membrane	13	0.00675959	0.58132464
	CC	GO:0005576~extracellular region	7	0.02323985	0.67349329
	CC	GO:0045121~membrane raft	3	0.03459211	0.67349329
	CC	GO:0070062~extracellular exosome	9	0.03710032	0.67349329
	CC	GO:0005615~extracellular space	6	0.03915659	0.67349329
	MF	GO:0008083~growth factor activity	4	0.00191792	0.13097688
	MF	GO:0005515~protein binding	22	0.0023814	0.13097688
	MF	GO:0005088~Ras guanyl-nucleotide exchange factor activity	3	0.01343914	0.31549128
	MF	GO:0003700~transcription factor activity, sequence-specific DNA binding	6	0.01432212	0.31549128
	MF	GO:0017124~SH3 domain binding	3	0.01434051	0.31549128
	MF	GO:0044212~transcription regulatory region DNA binding	3	0.04223068	0.55844233
	MF	GO:0043565~sequence-specific DNA binding	4	0.04427164	0.55844233
	MF	GO:0017147~Wnt-protein binding	2	0.04669972	0.55844233
	MF	GO:0017080~sodium channel regulator activity	2	0.04817069	0.55844233
	MF	GO:0001077~transcriptional activator activity, RNA polymerase II core promoter proximal region sequence-specific binding	3	0.05076748	0.55844233

**Table 2 vaccines-10-01557-t002:** KEGG pathway analysis of differentially expressed genes associated with mucoepidermoid and adenoid cystic carcinoma.

Salivary Gland Carcinoma	Term	Count	*p*-Value	Genes	FDR
Mucoepidermoid					
	hsa04512:ECM-receptor interaction	11	1.05 × 10^−13^	COMP, COL1A1, COL1A2, COL3A1, COL5A1, COL5A2, COL11A1, FN1, ITGB6, LAMB3, SPP1	3.98 × 10^−12^
	hsa04151:PI3K-Akt signaling pathway	13	3.29 × 10^−10^	COMP, COL1A1, COL1A2, COL3A1, COL5A1, COL5A2, COL11A1, FGF11, FGFR3, FN1, ITGB6, LAMB3, SPP1	6.25 × 10^−9^
	hsa04510:Focal adhesion	11	6.35 × 10^−10^	COMP, COL1A1, COL1A2, COL3A1, COL5A1, COL5A2, COL11A1, FN1, ITGB6, LAMB3, SPP1	8.04 × 10^−9^
	hsa04974:Protein digestion and absorption	6	1.07 × 10^−5^	COL1A1, COL1A2, COL3A1, COL5A1, COL5A2, COL11A1	8.15 × 10^−5^
	hsa04611:Platelet activation	6	7.11 × 10^−5^	COL1A1, COL1A2, COL3A1, COL5A1, COL5A2, COL11A1	4.50 × 10^−4^
	hsa05200:Pathways in cancer	7	0.00186707	WNT7B, FGF11, FGFR3, FN1, FZD10, LAMB3, MMP9	0.01013551
	hsa04310:Wnt signaling pathway	4	0.01174631	WNT7B, FZD10, SFRP2, SFRP	0.051564
	hsa04550:Signaling pathways regulating pluripotency of stem cells	4	0.01221253	WNT7B, FGFR3, FZD10, INHBA	0.051564
	hsa05205:Proteoglycans in cancer	4	0.0312654	WNT7B, FN1, FZD10, MMP9	0.11880852
	hsa04810:Regulation of actin cytoskeleton	4	0.03540698	FGF11, FGFR3, FN1, ITGB6	0.12231501
	hsa04810:Regulation of actin cytoskeleton	4	0.03540698	FGF11, FGFR3, FN1, ITGB6	0.12231501
Adenoid cystic					
	hsa05200:Pathways in cancer	10	7.75 × 10^−6^	RUNX1T1, WNT5A, BIRC3, EGF, FGF10, FGF12, FGF13, PLCB4, PRKACB, ZBTB16	6.83 × 10^−4^
	hsa04014:Ras signaling pathway	8	1.45 × 10^−13^	RASAL1, ANGPT1, CALML5, EGF, FGF10, FGF12, FGF13, PRKACB	6.83 × 10^−4^
	hsa05202:Transcriptional misregulation in cancer	7	2.88 × 10^−5^	CD14, ETV1, RUNX1T1, SIX1, FUT8, TMPRSS2, ZBTB16	7.11 × 10^−4^
	hsa04010:MAPK signaling pathway	8	3.02 × 10^−5^	CD14, CACNB2, EGF, FGF10, FGF12, FGF13, MAPT, PRKACB	7.11 × 10^−4^
	hsa04015:Rap1 signaling pathway	7	1.04 × 10^−4^	ANGPT1, CALML5, EGF, FGF10, FGF12, FGF13, PLCB4	0.00195539
	hsa04151:PI3K-Akt signaling pathway	8	2.16 × 10^−4^	ANGPT1, CREB3L1, CHAD, EGF, EIF4EBP1, FGF10, FGF12, FGF13	0.0033801
	hsa04922:Glucagon signaling pathway	5	4.73 × 10^−4^	PPARGC1A, CREB3L1, CALML5, PLCB4, PRKACB	0.00577649
	hsa04310:Wnt signaling pathway	4	0.01465669	WIF1, WNT5A, PLCB4, PRKACB	0.08610805
	hsa05221:Acute myeloid leukemia	3	0.01866771	RUNX1T1, EIF4EBP1, ZBTB16	0.09748695
	hsa04970:Salivary secretion	3	0.04135313	CALML5, PLCB4, PRKACB	0.15733407

**Table 3 vaccines-10-01557-t003:** Prognostic information of mucoepidermoid and adenoid cystic carcinomas hub candidate genes analyzed by KM plotter.

Genes	Mucoepidermoid Carcinoma	Adenoid Cystic Carcinoma
With significantly worse survival (*p* < 0.05)	FN1, SPP1	EGF, ERBB2
Without significantly worse survival (*p* > 0.05)	COL1A1, COL1A2, COL3A1, COL5A1, COL5A2, COL11A1, COMP, MMP9	PPARGC1A, WNT5A

## Data Availability

The supporting data for reported results can be found in Targeted RNA quantification of 770 genes in primary salivary gland carcinomas, DataSets for GSE153283, on line available in: https://www.ncbi.nlm.nih.gov/geo/query/acc.cgi?acc=GSE153283 (accessed on 31 May 2022).

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
