# Peer review of "Bioinformatic Analysis for Mucoepidermoid and Adenoid Cystic Carcinoma of Therapeutic Targets"

_vaccines, 2022, doi:10.3390/vaccines10091557_

Round 1
Reviewer 1 Report
The manuscript by Martínez-Ramírez et al used a bioinformatics approach to analyse gene expression in salivary gland neoplasms.
This analysis resulted in the identification of 4 genes (associated with poor survival) with potential therapeutic implications. Importantly, there are existing drugs against these molecules since they are also therapeutic targets in other neoplasms.
Comments
Some abbreviations are not required, eg in the results section it’s not necessary to abbreviate “biological process’, “cell components” or “molecular functions”. Similarly in the discussion, if a name is only used a couples of times (eg, cancer-associated fibroblasts) it’s better to use the full name to make it easier for the reader.
Lines 16 and 86, the word ‘posteriorly’ is unusual and may not be appropriate here. Maybe something like afterwards or subsequently is more suitable.
Line 108, please check if “DNA-templated” is a proper term.
Increase font in gene clusters (figure 1).
Author Response
Comments |
Action |
Some abbreviations are not required, eg in the results section it’s not necessary to abbreviate “biological process’, “cell components” or “molecular functions”. Similarly in the discussion, if a name is only used a couples of times (eg, cancer-associated fibroblasts) it’s better to use the full name to make it easier for the reader. |
Modified |
Lines 16 and 86, the word ‘posteriorly’ is unusual and may not be appropriate here. Maybe something like afterwards or subsequently is more suitable |
Modified |
Line 108, please check if “DNA-templated” is a proper term. |
The term is right, this is an abstracted correlation of DNA as template for transcription. |
Increase font in gene clusters (figure 1). |
The font and distribution of figure were modified |

Reviewer 2 Report
Known in the field based on previous literatures:
-
A neoplasm is a type of abnormal and excessive growth of tissue. Neoplasms may be benign (not cancer) or malignant (cancer).
-
Neoplastic tumors are often heterogeneous and contain more than one type of cell, including mucoepidermoid carcinoma (MECa) and adenoid cystic carcinoma (AdCC). MECa is the most common malignant salivary gland tumor, followed by AdCC.
-
The prime cause of disease comprises changes in the DNA sequence (mutation), copy number variations, and gain or loss of different genes.
In this manuscript authors reported following findings:
I have gone through the article titled “Bioinformatic analysis for mucoepidermoid and adenoid cystic carcinoma of therapeutic targets”. Authors scrutinized the genes using bioinformatics tool for both malignant salivary gland neoplasm- MECa and AdCC. They analyzed 32000 gene sets to identify the functional therapeutic targets of MECa and AdCC. Authors are analyzed and reported following findings-
-
GO analysis of differential expressed genes associated with MECa and AdCC and identified 97 and 86 upregulated gens for MECa and AdCC.
-
Authors analyzed the core genes using Kaplan-Meier plotter and interacting genes using STRING .
The data presented are interesting. There are, however, several issues that require the authors' attention. The following suggestions if incorporated could help in the better understanding of the significance of the work and implications.
Minor Concerns:
-
What is FOR in line 242?
-
There are many bioinformatic studies for MECa and AdCC available. Explain, how your study is different from rest excluding identification of two hub genes? Does it address a specific gap in the field?
- The discussion part is not sufficient and need to rewrite. Authors should compare and match their findings with previous studies. Authors should also discuss how many genes are matching with different experimental model studies and the possible outcome.
Author Response
Comments |
Action |
What is FOR in line 242? |
Modified, it was an editing error |
There are many bioinformatic studies for MECa and AdCC available. Explain, how your study is different from rest excluding identification of two hub genes? Does it address a specific gap in the field?
|
Modified. Information inserted in line 183 to 196. |
The discussion part is not sufficient and need to rewrite. Authors should compare and match their findings with previous studies. Authors should also discuss how many genes are matching with different experimental model studies and the possible outcome.
|
Modified. Information inserted in line 183 to 196, and in line 254 to 265. |
